# Fast Min-$\epsilon$ Segmented Regression using Constant-Time Segment Merging

**Ansgar Lößer** [1]   **Max Schlecht** [2]   **Florian Schintke** [3]   **Joel Witzke** [3]   **Matthias Weidlich** [2]   **Björn Scheuermann** [1]

## Abstract

Segmented regression is a statistical method that approximates a function $f$ by a piecewise function $\hat{f}$ using noisy data samples. *Min-$\epsilon$* approaches aim to reduce the regression function's mean squared error (MSE) for a given number of $k$ segments. An optimal solution for *min-$\epsilon$* segmented regression is found in $\mathcal{O}(n^2)$ time (Bai & Perron, 1998; Yamamoto & Perron, 2013) for $n$ samples. For large datasets, current heuristics improve time complexity to $\mathcal{O}(n \log n)$ (Acharya et al., 2016) but can result in large errors, especially when exactly $k$ segments are used. We present a method for *min-$\epsilon$* segmented regression that combines the scalability of top existing heuristic solutions with a statistical efficiency similar to the optimal solution. This is achieved by using a new method to merge an initial set of segments using precomputed matrices from samples, allowing both merging and error calculation in constant time. Our approach, using the same samples and parameter $k$, produces segments with up to $1{,}000\times$ lower MSE compared to Acharya et al. (2016) in about $100\times$ less runtime on datasets over $10^4$ samples.

## 1. Introduction

Segmented regression models the relation between a dependent (response) variable and a set of independent (predictor) variables by a piecewise function (Draper & Smith, 1981; Chen & Wang, 2009). It has applications in fields such as ecology (Shao & Campbell, 2002), econometrics (Yamamoto & Perron, 2013), clinical guidelines (Ansari et al., 2003), spatial gene analysis (Chitra et al., 2025) and computer science (Galakatos et al., 2019; Dai et al., 2020). Segmented regression estimates a piecewise function $\hat{f}$ based on noisy samples of a function $f$ denoting the true relationship between predictor and response variables. This involves specifying a segment for each interval of a partition of the independent variables. See Figure 1, which models 14 sample values using three segments. *Min-$\epsilon$* segmented regression sets a fixed number of segments $k$ and aims to construct these segments such that the estimation error is minimized.

A common instance of the problem is polynomial regression and the minimization of the mean-squared error (MSE) using ordinary least squares (OLS). For $k = 1$ and $n$ samples, it is a simple polynomial regression and takes $\mathcal{O}(n)$ time to solve. However, for $k \geq 2$, finding the positions of breakpoints is challenging. Most previous work relies on dynamic programming (Bai & Perron, 1998; Yamamoto & Perron, 2013). As shown by Acharya et al. (2016), finding the optimal solution with exactly $k$ segments requires $\mathcal{O}(n^2)$ time.

In practice, quadratic time complexity is increasingly problematic because of the growing amount of available data. Using a large number of samples is desirable to reduce noise impact. Therefore, heuristic methods that reduce time complexity to $\mathcal{O}(n \log n)$ have been proposed (Acharya et al., 2016). These methods create many initial segments and then merge them based on the minimal error of consecutive segments. However, the merging is greedy, either assuming knowledge on the noise distribution's variance, or being conducted separately for $\log(n)$ buckets over the segments' lengths. In either case, many more than the desired $k$ segments are created, which generally hampers a qualitative analysis of the breakpoint positions. Reducing the number of segments by post-processing in order to solve the actual segmented regression problem for a given $k$, in turn, increases the MSE by orders of magnitude.

In this paper, we present a new heuristic solution to the problem of *min-$\epsilon$* segmented regression that achieves near-optimal results and scales to large datasets, while using exactly $k$ segments. Our approach does not rely on any knowledge about the noise distribution or its variance. It follows the general idea of merging an initial set of segments, but we introduce a novel method: By leveraging precomputed matrices derived from the samples and the initial segment set, we achieve constant-time merging of two consecutive segments and computation of their combined error term.

---

[1]TU Darmstadt, Germany [2]Humboldt-Universität zu Berlin, Germany [3]Zuse Institute Berlin, Germany. Correspondence to: Ansgar Lößer <ansgar.loesser@kom.tu-darmstadt.de>.

*Proceedings of the $42^{nd}$ International Conference on Machine Learning*, Vancouver, Canada. PMLR 267, 2025. Copyright 2025 by the author(s).

In the remainder, after defining the problem (Sect. 2) and reviewing the state of the art (Sect. 3), we make the following contributions:

- We present a greedy approach for *min-$\epsilon$* segmented regression that is based on constant-time merging of segments and maintenance of their errors (Sect. 4).
- We analyse the complexity of our solution in terms of time and space. It requires $\mathcal{O}(n \log n)$ time and $\mathcal{O}(n)$ space for datasets with $n$ samples (Sect. 5).

We demonstrate the efficiency and effectiveness of our solution through a series of experiments (Sect. 6). For datasets exceeding $10^4$ samples, our technique runs about two orders of magnitude faster than best current methods. At the same time, accuracy is greatly improved, reducing the MSE by up to three orders of magnitude in comparison, resulting in an error averaging just 3 % above the optimal solution.

## 2. The Problem of Segmented Regression

For *general regression*, we are given $n \in \mathbb{N}$ observed samples of an underlying function $f : \mathbb{R}^d \to \mathbb{R}$. A sample $i \in \mathbb{N}, 1 \leq i \leq n$ is defined by a vector of $d$ independent variables $x_i \in \mathbb{R}^d$ and the dependent variable $y_i \in \mathbb{R}$. The *dependent* variables $y_i$ may suffer from noise or measurement errors. Errors in the *independent* variables $x_i$ are not considered here, as in many use cases, they are not affected by errors or the error is small enough to be negligible. A typical regression model is then defined by:

$$y_i = f(x_i) + \epsilon_i$$

The vector of error values $\vec{\epsilon} = (\epsilon_1, \ldots, \epsilon_n)^T$ is caused by noise in the measurement and is often considered to consist of independent, identically distributed (i.i.d.) Gaussian noise values. The goal of a regression is to find a function $\hat{f}$, that is as close as possible to $f$ and outputs a prediction $\hat{y}$ of the dependent variable.

A typical use case is polynomial regression. Given a sample at position $p$, the vector of independent variables for that sample can be defined as $x_i = (p^0, p^1, \ldots, p^{d-1})^T$, resulting in a regression by a polynomial of degree $d - 1$. For polynomials of degree one ($d = 2$), this is called *linear regression*. While all methods in this paper work with an arbitrary vector of independent values, we will present polynomial regression for the sake of simplicity.

For *segmented* regression, $\hat{f}$ is a $k$-piecewise function, which is represented by $k$ individual segments. Given one strict order for all samples (e.g., the x-coordinate for polynomial regression), every segment $j$ is defined by its own mathematical function $f_j$ where $1 \leq j \leq k$, and the starting point of the segment. At the borders between two adjacent segments (called breakpoints), the predicted value of the

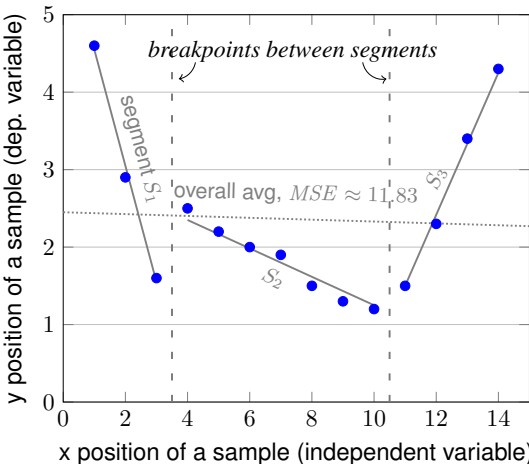

*Figure 1.* Illustration of segmented regression: For 14 samples (blue data points), three segments ($S_1$, $S_2$, $S_3$, gray lines) are constructed, separated by breakpoints.

two corresponding functions does not need to be equal, i.e., the overall piecewise function is not necessarily continuous.

In *min-$\epsilon$ segmented* regression, a fixed number of segments $k$ is given. The algorithm finds functions $\hat{f}_j, 1 \leq j \leq k$ that minimize an error metric describing the distance between $y$ and $\hat{y}$ (Chen & Wang, 2009). The solution should be as close as possible to the function $f$ in order to accurately model the underlying correlation between $x$ and $y$.

A common general estimator for regressions is ordinary least squares (OLS). This estimator minimizes the mean-squared error (MSE) for a sampled dataset:

$$MSE = \frac{1}{n} \sum_{i=1}^{n} (\hat{y}_i - y_i)^2 = \frac{1}{n} \sum_{i=1}^{n} (\hat{f}(x_i) - y_i)^2$$

According to the Gauss-Markov theorem, OLS is the estimator with the minimum variance of all estimators that are linear combinations of the samples' independent variables, as long as the errors in $\epsilon$ are uncorrelated, have a mean of zero and a constant variance $\sigma$ (Hallin, 2014). This results in a maximum likelihood estimation (MLE) for a Gaussian error distribution. While these properties do not necessarily apply to other distributions, OLS is, nevertheless, often used for other or unknown error distributions, making it one of the most common regression methods.

For a function with $k = 1$, e.g., a simple polynomial regression, the OLS can be computed analytically in $\mathcal{O}(n)$ time. However, for a piecewise function with $k \geq 2$, the problem is to find good positions for the breakpoints. The number of segmentation combinations resembles the composition of an integer in combinatorics. In our case, we want to compose the integer $n$ with exactly $k$ parts. The number of compositions is given by the binomial coefficient $\binom{n-1}{k-1}$. If every

combination is tried, the OLS would need to be computed for every potential solution, so that this solution scales like $\Theta\left(n^k\right)$. Even if the number of segments is small, for $k > 1$, computing all combinations becomes soon intractable for a growing number of samples. Since more samples result in more accurate regression models by minimizing the effect of the noise, faster algorithms are desirable.

## 3. State-of-the-art Segmented Regression

Solving the *min-ϵ* segmented regression problem often relies heavily on dynamic programming, according to several studies (Bai & Perron, 1998; Yamamoto & Perron, 2013). The algorithm is considered folklore and is explained in Acharya et al. (2016). It fills a table with optimal regression solutions for $1..k$ segments for the first $1..n$ samples. Finding the best solution for one entry takes $\mathcal{O}(n \cdot d^2)$ time, assuming solutions for smaller $k$ and $n$ are already computed. Incrementally filling this table takes $\mathcal{O}(k \cdot n^2 \cdot d^2)$ time, scaling quadratically with the number of samples.

This method is more feasible than trying all combinations naïvely. It remains optimal in minimizing the global MSE for the sampled dataset and can be considered the OLS solution in segmented regression. However, an $\mathcal{O}(n^2)$ runtime is prohibitive for large datasets. More samples, often noisy but cheap or free to collect, can increase regression accuracy, as confirmed by our evaluation results in Sect. 6.

Since more samples lead to more accurate regression models by minimizing noise effects, faster, not necessarily optimal, algorithms are desirable. *Heuristic approaches* for *min-ϵ* segmented regression, such as those by Acharya et al. (2016) achieve this with a time complexity of $\mathcal{O}(n \log n)$. These methods start with some initial segments and merge them later either based on knowledge about $\sigma$, the noise's variance, or based on the error for segments grouped by their length. Although the practical statistical efficiency of the latter approach is unclear, regression functions derived from the former approach tend to have high error rates that do not substantially decrease with additional samples. To alleviate this issue, methods creating $2k$ or $4k$ piecewise functions are used. This decreases prediction errors with more segments but restricts qualitative analysis, such as determining where the underlying function actually changes.

Experiments by Acharya et al. (2016) show that adding more segments reduces error, but it remains two to four times larger than the optimal solution's error. Compensating this effect by using more data is possible, but also requires more compute time. In this trade-off between compute time and accuracy the state-of-the-art heuristics often outperform the exact algorithms, but this is only feasible if sufficient samples are available and the specific usecase allows for the creation of more than $k$ segments.

## 4. Novel Greedy Approach

To accurately approximate segment positions and their regression models, we developed a greedy algorithm. Minimizing the global MSE of our regression function $\hat{f}$ is equivalent to minimizing the residual sum of squares (RSS), which means minimizing the RSS for all $s$ segments together:

$$RSS = MSE \cdot n = \sum_{i=1}^{n} (y_i - \hat{y}_i)^2 = \sum_{j=1}^{s} RSS_j$$

We start by creating as many segments as possible and calculate the individual regression functions using OLS. We also compute the potential additional merge error between neighboring segments. Then, we iteratively merge neighboring segments that increase the RSS the least (see Figure 2 for an exemplary illustration). Once two segments are merged, we recalculate the merge errors of the surrounding segments to decide which segment should be merged next with its neighbor. This process involves regenerating the model for the samples of those segments.

Using the typical approach to calculate the OLS, the time to compute this model generation scales linearly with the number of samples in the segment. The number of merge operations to reach $k$ segments also scales linearly with the number of initial segments—and therefore with the number of samples—resulting in a naïve time complexity of $\mathcal{O}(n^2)$ for $n$ samples.

The heuristic approach of Acharya et al. (2016) employs a similar greedy approach. They reduce time complexity by using multiple methods to reduce model recalculations. This

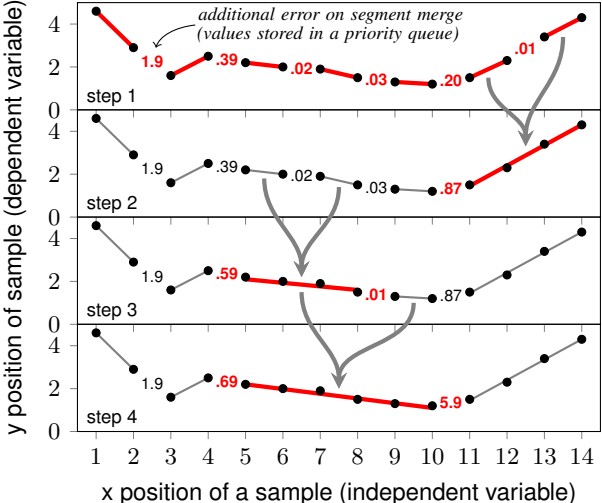

*Figure 2.* Exemplary iterative merging of the segments for the solution shown in Figure 1 with our algorithm, where the least-possible additional error is added. Bold red highlights changes in each step.

includes starting with fewer initially placed segments and delaying recalculations until an error threshold is reached. These methods improve performance but reduce the achievable accuracy and require estimating sampling noise $\sigma$. In contrast, we store segments in a way that allows constant time merging.

### 4.1. Greedy Segmented Regression

The algorithm has three main phases: (1) place as many segments as possible and precalculate some attributes per segment, (2) reduce the number of segments to the target $k$ by merging those that cause the smallest increase in overall RSS. (3) execute a post-optimization step to compensate for suboptimal initial segment placement.

**Initial placement.** Initially, as many segments as possible are placed. Without overlapping segments, a maximum of $\lfloor \frac{n}{d} \rfloor$ segments can be placed (e.g., linear regression means $d = 2$), as at least $d$ samples per segment are needed to compute a uniquely defined OLS regression (see Sect. 4.2).

For each placed segment, an optimal model is generated using OLS. Except for the last segment, the error for each segment is zero, since $d$ points can be perfectly matched by exactly one specific polynomial of degree $d - 1$. It is now possible to calculate a merge cost $M(S_j)$ for each segment $S_j$. This cost metric indicates how much the overall RSS will increase if the segment is merged with its successor $S_{j+1}$, defined as follows:

$$M(S_j) = RSS(S_j \cup S_{j+1}) - (RSS(S_j) + RSS(S_{j+1}))$$

This metric is never negative because combining segments cannot improve model performance or decrease total error. To quickly find which segment to merge next, we store the metric and its segment reference in a priority queue.

**Segment reduction.** We keep taking the segment with the lowest merge cost from the queue and merge it with the next segment until we have the desired number of $k$ segments. For each merge, this operation removes the original segments and adds a new merged one. It also shifts the indices of all following segments down by one.

Merging two segments induces local changes in $M$. To draw the correct segment in the next iteration, we update the cost of the new segment and its predecessor in the priority queue after each merge operation (bold red numbers in Figure 2).

**Placement optimization.** Step 5 in Figure 2 would lead to segment $S_2$ being merged into $S_3$. Regardless of the specific last merge step, it is impossible to separate the two points of $S_2$, as segments are never divided. To mitigate any negative impacts, we fine-tune breakpoint positions after reducing to $k$ segments. Each breakpoint position can be

adjusted by a quarter of an adjacent segment's size. Within this range, all possible positions are evaluated, and the one with the smallest RSS determines the final placement.

Optimizing one breakpoint influences the local models of surrounding segments. Larger segments are considered to be more stable because removing the first or last quarter of their samples typically has a smaller effect on the model. Therefore, we start with the breakpoints adjacent to the largest segments and continue in descending order.

### 4.2. Merging Segments in Constant Time

To show how we merge segments in constant time, we first look at the calculation of a single segment's OLS. Calculating the OLS of $n$ samples takes linear time. Given a matrix $X$ for independent variables of the samples and the corresponding vector of dependent variables $\vec{y}$, the vector with regression parameters $\beta$ for a polynomial of degree $d - 1$ can be calculated as follows.

$$X = \begin{bmatrix} x_1^0 & x_1^1 & x_1^2 & \cdots & x_1^{d-1} \\ x_2^0 & x_2^1 & x_2^2 & \cdots & x_2^{d-1} \\ x_3^0 & x_3^1 & x_3^2 & \cdots & x_3^{d-1} \\ \vdots & \vdots & \vdots & \ddots & \vdots \\ x_n^0 & x_n^1 & x_n^2 & \cdots & x_n^{d-1} \end{bmatrix}, \beta = \left( X^T X \right)^{-1} X^T \vec{y}$$

Since $X$ is a Vandermonde matrix, the Gram matrix $X^T X$ is positive semidefinite and is invertible if $X$ contains at least $d$ distinct input samples ($n \geq d$). Given the vector $\beta$, this regression model can be used to predict a value $\hat{y}_i$ for the $i$-th sample. Here, $X_i$ is the matrix $X$, but only containing the independent variables vector of the $i$-th sample. It is also possible to calculate the minimized mean squared error.

$$\hat{y}_i = \hat{f}(x_i) = X_i \beta$$

$$RSS = \sum_{i=1}^{n} \left( \hat{f}(X_i) - y_i \right)^2 = |X\beta - \vec{y}|^2$$

$$MSE = \frac{RSS}{n} = \frac{|X\beta - \vec{y}|^2}{n}$$

We introduce two additional matrices: $A$ of dimension $d \times d$, and $B$ of dimension $d \times 1$. These matrices can be used to calculate the OLS in exactly the same way as shown above.

$$A = X^T X, \quad B = X^T \vec{y}, \quad \beta = A^{-1} B$$

We can store a segment with the pre-computed matrices $A$ and $B$ with constant storage, independent of the number of points $n$. Assuming we have two segments with two distinct sets of samples $S_u$ and $S_v$, we can calculate the corresponding matrices. Furthermore, we can compute the matrices of a segment with the joint set by adding the matrices.

$$A_{u \cup v} = A_u + A_v, \quad B_{u \cup v} = B_u + B_v$$

Merging two segments with these pre-computed matrices is an operation of $\Theta(d^2)$ time and will not scale with the number of samples. By subtracting instead of adding the matrices, it is also possible to remove a subset of samples from a segment. $A$ and $B$ can also be calculated for single samples. While it is not possible to calculate the inverse of $A$ and derive the model parameters $\beta$ if $d > 1$ for individual samples, it is a valid strategy to be able to add single samples to a segment or remove individual samples. This is beneficial in the placement optimization step.

### 4.3. Calculating the Error of a Segment

In our algorithm, after merging, we need to compute the error of the resulting segment to update the merge cost metric. To avoid iterating over every sample, we demonstrate how to derive the RSS in constant time.

In this section, we only consider samples inside the current segment, where $n$ is the number of samples inside that segment. We introduce two new square matrices, $C$ and $D$, both of dimension $d + 1$. $C$ can be defined for a subset of samples, similar to $X$, $A$ and $B$. $C_i$ is the matrix $C$ for a single sample. The matrix $C_i$ only depends on the values in $A_i$ and $B_i$ and the value $y_i^2$ for the $i$-th sample. $D$ can be calculated using the regression parameters $\beta$. By transposing and simplifying the equation, where '$\odot$' is the Hadamard product (element-wise product of two matrices of same size), we can show[1] that:

$$C_i = \begin{bmatrix} A_i & B_i^T \\ B_i & y_i^2 \end{bmatrix}, \quad D = \begin{bmatrix} \beta \\ -1 \end{bmatrix} \cdot \begin{bmatrix} \beta \\ -1 \end{bmatrix}^T$$

$$RSS_i = (X_i\beta - y_i)^2 \qquad \Big| \text{ expand } X_i\beta$$

$$= \sum_{\ell,m=1}^{d+1} (D \odot C_i)_{\ell,m}$$

$$RSS = \sum_{i=1}^{n} RSS_i = \sum_{i=1}^{n} \sum_{\ell,m=1}^{d+1} (D \odot C_i)_{\ell m} \Big| C = \sum_{i=1}^{n} C_i$$

$$= \sum_{\ell,m=1}^{d+1} (D \odot C)_{\ell m}$$

The error of a segment is computed by the grand sum of the element-wise multiplication of the products of regression parameters $D$ and the products of all dependent and independent variables summed for all samples $C$. All described properties of matrices $A$ and $B$ also apply for $C$: $C$ is a square symmetric matrix that can be combined for multiple sample sets by addition. As matrices $A$ and $B$ are submatrices of $C$, only matrix $C$ has to be stored for every segment, along with the size or position of the segment.

––––––––––––––––––––

[1]A detailed derivation is shown in the appendix.

Combining segments is still possible in $\Theta(d^2)$ time by adding the matrices $C_u$ and $C_v$ for two segments $u$ and $v$. To calculate the error, we need to calculate $D$ and, therefore, need the model parameters. This involves calculating the inverse of $A$ or solving the linear system of $\beta = A^{-1} \cdot B$, which is typically done in $\mathcal{O}(d^3)$ time. All of these operations are independent of the number of points, so for a typical regression algorithm with a constant number of dimensions we are able to merge two segments and calculate the resulting RSS in constant time.

The cost of merging two segments can be further reduced. For high values of $d$, there are more efficient ways of calculating $\beta$. If just a single point is added to a segment, it is possible to use the Sherman-Morrison formula to do a rank-1 update of the already known matrix inverse in $\mathcal{O}(d^2)$ time. However, since our algorithm often merges segments much larger than a single point and the number of dimensions is often limited—e.g., $d = 2$ for segmented linear regression—we consider this to be a minor optimization and not relevant for the evaluation of our algorithm.

## 5. Complexity Analysis

In this section, we study the time and memory complexity of our algorithm with a focus on the influence of the number of samples $n$. We also consider how the number of dimensions $d$ affects these metrics. However, since $n \gg d$ must hold true, and $d$ is often a chosen constant for a specific regression model, it will not be the focus of our research.

The data structures to store segments have a great impact on the complexity of our algorithm. We use a double-linked list to quickly find the predecessor and successor of a segment and delete merged segments. Additionally, we implement a priority queue using a binary heap with reference tracking, which quickly identifies the next segment for merging and updates the merge costs of neighboring segments.

### 5.1. Memory Complexity

As described in Sect. 4, our algorithm first creates $\lfloor \frac{n}{d} \rfloor$ individual segments. Afterwards, the number of segments is continuously reduced until only $k$ segments are left.

For every segment, we need to store the summed-up matrix $C$. As shown in Sect. 4.3, this matrix is a squared symmetric matrix of dimension $d+1$. It can be stored with $\lceil \frac{1}{2}(d+1)^2 \rceil$ values. We need to store the size or position of a segment and the current merge cost, both with fixed sizes. Also, we need to reference all segments in a heap and a linked list. The size of a linked list grows linearly with the number of elements. While the overhead of a typical heap is constant, the reference tracking adds constant overhead per entry. The overall memory overhead of these data structures scales linearly with the number of elements. Hence, it is equivalent

to a constant overhead per segment.

Since we know the maximum number of segments and how the amount of data scales for a single segment, we can conclude that the memory complexity is given by:

$$\mathcal{O}\left(\left\lfloor \frac{n}{d} \right\rfloor \cdot \left\lceil \frac{(d+1)^2}{2} \right\rceil\right) = \mathcal{O}\left(\frac{n}{d} \cdot d^2\right) = \mathcal{O}(n \cdot d)$$

The amount of memory needed scales linearly with the number of data points $n$. If all samples need to fit into memory anyway, this can be considered optimal.

## 5.2. Time Complexity

We analyze the time complexity for the three phases of our algorithm individually.

**Initial placement.** First, $\lfloor \frac{n}{d} \rfloor$ segments are placed and inserted into the linked list. For every segment, we calculate the merge error cost with its successor, resulting in $2 \cdot \lfloor \frac{n}{d} \rfloor$ model generations, with a time complexity of $\mathcal{O}(d^3)$. All merge costs need to be inserted into the binary heap. This would normally result in a time complexity of $\mathcal{O}(s \log s)$ for $s$ segments. However, the initial insertion of many elements into a binary heap—often called *heapify*—can be optimized to run in amortized complexity of $\mathcal{O}(s)$. The resulting time complexity for the initial phase is defined by:

$$\mathcal{O}\left(2 \cdot \left\lfloor \frac{n}{d} \right\rfloor \cdot d^3 + \left\lfloor \frac{n}{d} \right\rfloor\right) = \mathcal{O}(n \cdot d^2)$$

**Segment reduction.** To reduce the number of segments to $k$, we perform $\lfloor \frac{n}{d} \rfloor - k$ merge-iterations. Since $k$ is considered to be a small constant, we perform $\mathcal{O}(\lfloor \frac{n}{d} \rfloor)$ iterations. In every iteration, we take an element from the heap ($\mathcal{O}(\log s)$), remove that element from the linked list ($\mathcal{O}(1)$), generate the new models ($\mathcal{O}(d^3)$) and update the values in the heap ($\mathcal{O}(\log s)$). The overall time complexity is:

$$\mathcal{O}\left(\left\lfloor \frac{n}{d} \right\rfloor \cdot \left(\log \left\lfloor \frac{n}{d} \right\rfloor + d^3\right)\right) = \mathcal{O}(n \cdot (\log n + d^2))$$

**Placement optimization.** To optimize the determined segments, we vary the positions of the breakpoints. Moving a breakpoint by one sample is similar to removing and adding a segment of size one. By storing and updating $A^{-1}$ with the Sherman-Morrison formula, this can be done in $\mathcal{O}(d^2)$. This operation is done for $\frac{n}{2}$ samples, resulting in a time complexity of $\mathcal{O}(n \cdot d^2)$.

Thus, the overall time complexity of our algorithm is given by the maximum of the three steps above:

$$\mathcal{O}(n \log n + n \cdot d^2)$$

Considering the number of samples $n$, the runtime of our algorithm scales with $\mathcal{O}(n \log n)$.

# 6. Evaluation

We use real and synthetic data to compare the accuracy and the runtime of our algorithm with the established approaches. The best exact solution, *dynamic program* (DP), is compared to our approach and the state-of-the-art heuristic (Acharya et al., 2016). Table 1 shows the algorithms' properties.

*Table 1.* Qualitative Comparison of Approaches

| Feature | DP | Our Appr. | Acharya | | |
| --- | --- | --- | --- | --- | --- |
| | | | $k$ | $2k$ | $4k$ |
| Correct $k$ | ✓ | ✓ | ✓ | ✗ | ✗ |
| Qualitative BP | ✓ | ✓ | ✗ | ✗ | ✗ |
| Exact | ✓ | ✗ | ✗ | ✗ | ✗ |
| Memory compl. | $n$ | $n$ | $n$ | $n$ | $n$ |
| Time compl. | $n^2$ | $n \log n$ | $n \log n$ | | |
| Rel. MSE (§6.1) | 1.00 | 1.03 | 495.13 | 3.43 | 4.95 |

Acharya's method places $k$, $2k$ or $4k$ segments. While the first version places the intended $k$ segments, it fails to accurately find the *breakpoint positions* (BP), where changes in $f$ occur, with real data (Sect. 6.2), leading to high error in the regression results. Placing more segments drastically reduces the error for synthtetic data but is still less effective than our approach (Sect. 6.1).

The dataset generation and measurement code use Python. The optimal dynamic program and heuristic were taken from Acharya et al. (2016). For exact control of data structures, we implemented our approach in C++. While programming language and implementation details might influence runtime, they should *not* change the results by orders of magnitude in this case. According to Acharya et al. (2016), their operations run similarly fast in C and Julia.

All experiments were done on an *AMD Ryzen Threadripper PRO 5955WX* system with Ubuntu 24.04 LTS. Details, measurement data, source code, experiment setup, our analysis pipeline, and an additional evaluation regarding parameter $d$ are included in the supplemental material of this paper[2].

## 6.1. Synthetic Data

Akin to Acharya et al. (2016), we generate piecewise continuous function with $k = 6$ segments for evaluation at $n$ points. Five random positions are chosen as segment start points between 0 and $n$. The first segment starts at 0, and the last ends at $n$. At each joint, we randomly select a $y$ value between 0 and 1. Segments interpolate linearly from their start to end points.

We take evenly spaced points from a function, add Gaussian noise using two fixed standard deviation ($\sigma$) values, and measure the time needed for a $k$-segmented regression. We

---

[2]The supplemental material is available at https://github.com/Loesgar/mvsr/tree/paper-icml-25.

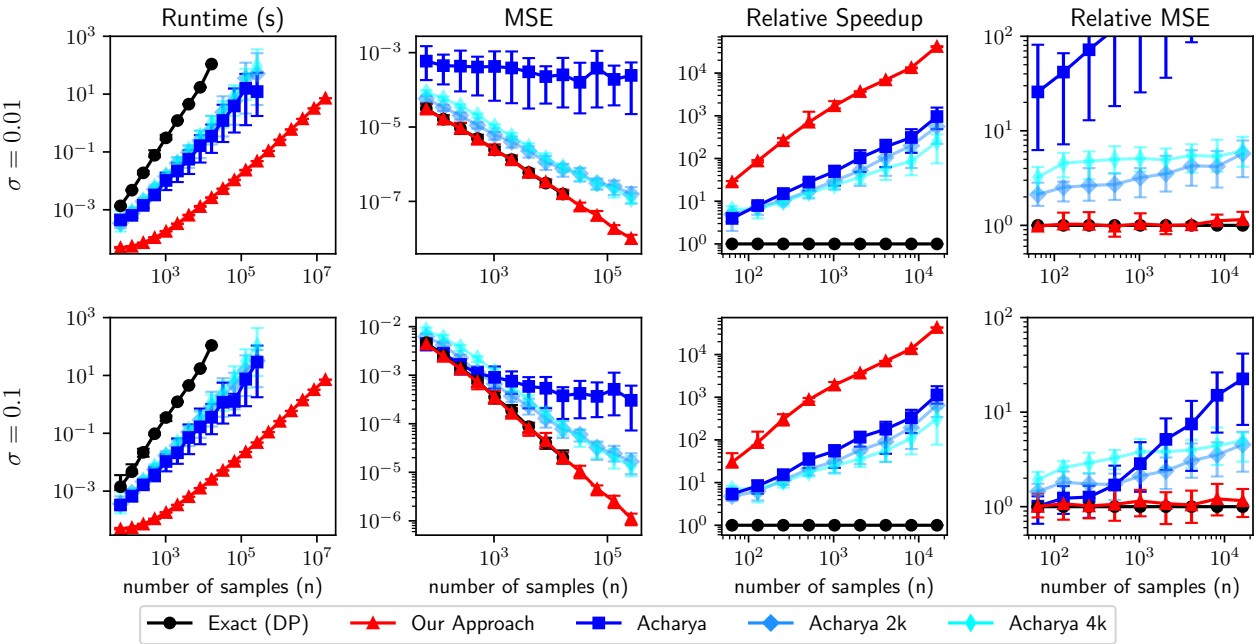

*Figure 3.* Evaluation results (log-log scale) from synthetic data with Gaussian noise. For each point, 100 functions $f$ with $k = 6$ segments were tested across all regression algorithms. Data points show average measurements. Due to excessive runtime, we limited maximum sample size of the other approaches. Error bars mark the interquartile range (50 % of all values). MSE measures error between predicted $\hat{y}$ and the actual value $f(x)$. Relative figures compare individual results to those from the exact dynamic program on the same function.

calculate the MSE using the true values from the original function, not the noisy data. This eliminates the impact of high noise and shows how error decreases as more samples are used in the regression.

Figure 3 shows performance results for $\sigma$ values of 0.01 and 0.1 (equivalent to 1 % and 10 % of the value range). The correct $\sigma$ was also used as an input parameter for the competing heuristic, which our algorithm did not require.

The results are similar to those of Acharya et al. (2016). They reported an MSE of 2 to 4 times higher than the dynamic program. We observed a range of 2 to 5 times, the values and trends matching theirs, confirming their findings. Additionally, we validated that the $1k$ approach's accuracy does not improve with more samples. Our approach consistently outperforms existing heuristics in both runtime and MSE, even those using more segments. Our approach's runtime is consistent for a given number of inputs with little variation. The linear appearance on a log-log graph suggests initial slowdown might be from measurement inaccuracy and overhead, not the actual performance.

Our approach sometimes outperforms the exact dynamic program. The exact algorithm minimizes error based on sampled data, so it is possible to find a solution that is similar but slightly less accurate to the sampled data, yet closer to the true function $f$. This often happens in high-noise scenarios.

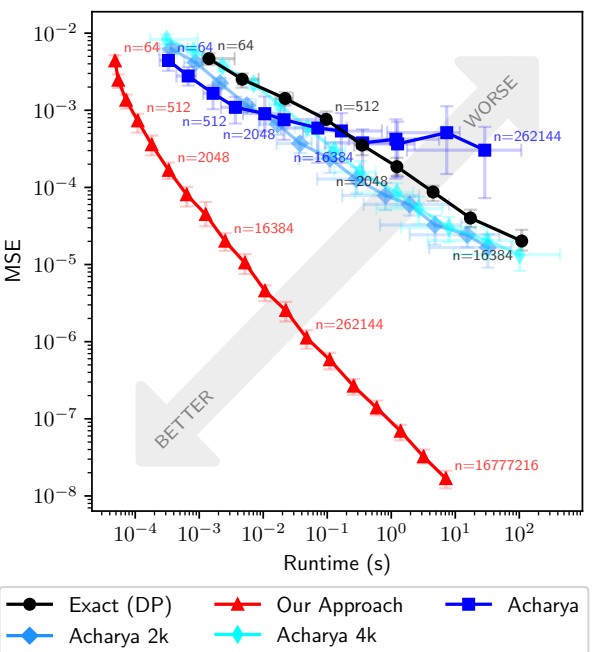

*Figure 4.* By varying the sample size $n$, we can compare each algorithm's accuracy within a fixed computation time (based on results of Figure 3 with $\sigma = 0.1$). Our algorithm outperforms all evaluated solutions, especially if many samples are available.

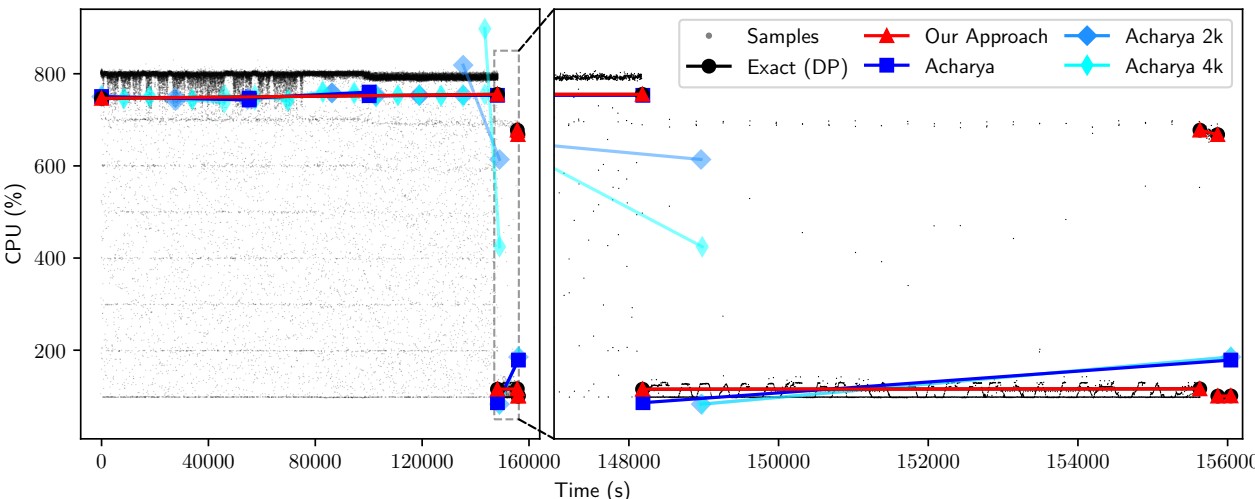

*Figure 5.* Segmented regression results on real data. The data show samples from a timeseries, the CPU usage of an execution of the common scientific program 'bwa' of the scientific workflow 'sarek'. The first segment is especially large, followed by multiple short periods of different behavior. The later part, starting at roughly 148,000s is magnified on the right. The optimal solution (DP) is equal to our approach (red).

We analyzed the tradeoff between compute time and MSE for $\sigma = 0.1$ by varying the number of input samples $n$, ensuring competing algorithms had similar average runtime. The results are shown in Figure 4. Our approach has nearly optimal error with much faster computation, outperforms all competitors, even those using more segments.

Our method is slightly less accurate than the exact dynamic program if the computing power is not taken into account. The exact method usually performs better in this situation. On average, with a noise level of $\sigma = 0.01$, our approach only increased the MSE by 3 % (see Table 1).

### 6.2. Practical Data

To evaluate our algorithm, we used time series data showing CPU usage during the 43-hour execution of the *bwa* program (Li, 2013) from *sarek* workflow (Garcia et al., 2020). Every two seconds a sample was measured, resulting in 70,607 samples. The dataset shows roughly four execution stages with different behavior. Unlike synthetic data, it is diverse: noise is not normally distributed, variance is high and varies between segments. Most samples belong to the first stage, while all other stages share fewer samples.

Figure 5 shows the dataset and the regression functions $\hat{f}$ using different algorithms. Our approach matches the optimal solution from DP exactly, clearly identifying the execution phases. The competing heuristic *Acharya* fails to detect correct breakpoints, places unnecessary segments inside the first execution phase. Since the exact ground truth is unknown, we cannot calculate MSE against it. Instead, we calculated MSE between sampled data and predictions.

*Table 2.* Evaluation results for the dataset shown in Figure 5.

| Attribute | DP | Our Appr. | Acharya $k$ | 2$k$ | 4$k$ |
|---|---|---|---|---|---|
| Runtime (s) | 2938.33 | **0.01** | 0.30 | 0.33 | 0.34 |
| MSE ($\times 10^4$) | **1.82** | **1.82** | 1.87 | 2.05 | 1.99 |
| Rel. Runtime | $\times 2.9 \cdot 10^5$ | $\times\mathbf{1}$ | $\times 30$ | $\times 33$ | $\times 34$ |
| Rel. Error (+%) | **0.00** | **0.00** | 2.61 | 12.20 | 9.18 |

Table 2 shows that using Acharya does not match our method or DP. While DP gives an accurate result, it needs more than forty minutes. Our approach finds the exact same solution in a few milliseconds.

This specific scenario shows that more segments not necessarily yield more accurate results. Acharya performed worse when placing $2k$ or $4k$ segments. Only with exactly $k$ segments Acharya created a breakpoint that was very close to a breakpoint of DP and our approach.

## 7. Related Work

The Python module 'piecewise-regression' and the R package 'segmented' use an algorithm by Muggeo for fitting line segments *and* breakpoints in *continuous* piecewise regression functions (Pilgrim, 2021; Muggeo, 2008; 2003). The 'pwlf' Python module optimizes breakpoint locations with either a differential evolution algorithm or a faster gradient based method (Jekel & Venter, 2019). However, due to the focus on continuity, their performance is worse than our proposed algorithm, which is more reliable and can handle non-continuous data. Our algorithm effectively identifies segment positions for piecewise continuous functions and

can be adapted to create continuous piecewise functions from the segmentation results. The effectiveness of this adaptation compared to existing approaches is beyond the scope of this paper.

The algorithm from Acharya et al. (2016), compared above, was later extended by Diakonikolas et al. (2020) to segment along multidimensional breakpoints (e.g. place rectangles). They claim it to be the first efficient multidimensional segmented regression algorithm that works in any fixed dimension. Our algorithm might also be extended for arbitrary dimensions similarly. Evaluating its performance in a multidimensional setting is open for future research.

The related *min-#* problem involves finding a function $\hat{f}$ close enough to another within an error limit $\epsilon$ while using the fewest segments (Chen & Wang, 2009). It is easier than related problems, with solutions that run in $\mathcal{O}(n)$ time (Tomek, 1974; Imai & Iri, 1987; Neubauer, 2009). Solving *min-#* can often be extended to solving min-$\epsilon$, but this increases complexity to about $\mathcal{O}(n \log n)$.

Recent works by Warwicker & Rebennack (2023; 2024) focus on solving the *min-#* problem with strict continuity in regression functions. Some of their algorithms only handle continuous functions, not discrete data points, which limits comparisons to our work.

## 8. Conclusion

We presented a new heuristic to solve the *min-ε* segmented regression problem. The greedy algorithm merges neighboring segments in constant time. Our algorithm finds a solution using exactly $k$ segments without needing any specific information about the input. We showed that it can analyse $n$ samples in a runtime of $\mathcal{O}(n \log n)$ and has a memory complexity of $\mathcal{O}(n)$.

We evaluated the algorithm's speed and accuracy against state-of-the-art heuristic and optimal solutions. The resulting regressions are close to the optimum, outperforming all other presented heuristics, while our algorithm is orders of magnitudes faster. The algorithm provided the best balance of speed and accuracy compared to others. Our algorithm is usually the best option. However, if there are only a few samples and very large computing resources, the competing dynamic program is, on average, slightly more accurate.

## Acknowledgements

This work received funding from the German Research Foundation (DFG), CRC 1404: *FONDA: Foundations of Workflows for Large-Scale Scientific Data Analysis*.

## Impact Statement

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

# Appendix

This document represents the technical appendix of the the paper *Fast Min-$\epsilon$ Segmented Regression using Constant-Time Segment Merging*. The main contribution is the clarification and better explanation of the mathematical formulas in Section 4 of the paper and an experimental evaluation of the influence of the parameter $d$ on the computation time and accuracy of our algorithm.

## A. Constant-Time Segment Merging

This section describes the mathematical formulas from Section 4. We will use the same connotations and names as in the paper without further explanation.

### A.1. Segment Merging

The paper presented multiple matrices $A$, $B$, and $C$ based on $X$ (the matrix containing the independent variables of a sample for polynomial regression) and $\vec{y}$ (the vector containing the dependent variables of the samples). It is stated that these matrices can be calculated for individual samples or sets of samples and can later be added together to get the matrices of the combined sample set. Since $A$ and $B$ are submatrices of $C$, we will focus on showing this property for $C$.

The paper defines the matrix $C$ the following way:

$$X = \begin{bmatrix} x_1^0 & x_1^1 & x_1^2 & \dots & x_1^{d-1} \\ x_2^0 & x_2^1 & x_2^2 & \dots & x_2^{d-1} \\ x_3^0 & x_3^1 & x_3^2 & \dots & x_3^{d-1} \\ \vdots & \vdots & \vdots & \ddots & \vdots \\ x_n^0 & x_n^1 & x_n^2 & \dots & x_n^{d-1} \end{bmatrix}$$

$$A = X^T X, \quad B = X^T \vec{y}$$

$$C = \begin{bmatrix} A & B^T \\ B & \sum_{i=1}^n y_i^2 \end{bmatrix}$$

Another way to represent $C$ is by introducing a new matrix $X^+$, which is similar to $X$ but contains all information of the samples (including the dependent variable).

$$X^+ = \begin{bmatrix} X & \vec{y} \end{bmatrix} = \begin{bmatrix} x_1^0 & x_1^1 & x_1^2 & \dots & x_1^{d-1} & y_1 \\ x_2^0 & x_2^1 & x_2^2 & \dots & x_2^{d-1} & y_2 \\ x_3^0 & x_3^1 & x_3^2 & \dots & x_3^{d-1} & y_3 \\ \vdots & \vdots & \vdots & \ddots & \vdots & \vdots \\ x_n^0 & x_n^1 & x_n^2 & \dots & x_n^{d-1} & y_n \end{bmatrix}$$

The matrix $C$ can now easily be calculated from $X^+$, since $C = {X^+}^T X^+$. Any single element in the resulting matrix $C$ is the sum of the element-wise products of two columns in $X^+$. The following equation shows this for the value $C_{1,2}$.

$$C_{1,2} = x_1^0 \cdot x_1^1 + x_2^0 \cdot x_2^1 + x_3^0 \cdot x_3^1 + \dots + x_n^0 \cdot x_n^1$$

$$C_{1,2} = \sum_{i=1}^n (x_i^0 \cdot x_i^1)$$

I.e., a single value in the matrix $C$ is the sum of a product of two values of a single sample, for every sample. There is never a product of two values that depend on two different samples. Adding more samples only results in longer columns in $X^+$. Calculating $C_u$ on a subset $u$ of the samples and adding it later to $C_v$ containing the rest of the samples is exactly the same as calculating $C$ for all samples in the first place. The only thing that might change is the order of the additions. This is irrelevant since addition is a commutative operation.

### A.2. Error Calculation

We now show how to use the matrix $C$ to calculate the RSS. The paper already shows how the OLS paramter vector $\beta$ can be calculated using the submatrices $A$ and $B$. The RSS for the sample $i$ can be calculated the following way:

$$RSS_i = (\hat{f}(x_i) - y_i)^2 = (X_i\beta - y_i)^2 \quad \Big| \text{ expand } X_i\beta$$

$$= (x_i^0\beta_1 + x_i^1\beta_2 + \dots + x_i^{d-1}\beta_d - y_i)^2$$

This can be further expanded, resulting in the following term:

$$\begin{array}{ccccccccc}
 & x_i^0\beta_1 & \cdot & x_i^0\beta_1 & + & \dots & - & x_i^0\beta_1 & \cdot & y_i \\
+ & x_i^1\beta_2 & \cdot & x_i^0\beta_1 & + & \dots & - & x_i^1\beta_2 & \cdot & y_i \\
\vdots & & & & & & & & & \\
+ & x_i^{d-1}\beta_d & \cdot & x_i^0\beta_1 & + & \dots & - & x_i^{d-1}\beta_d & \cdot & y_i \\
- & y_i & \cdot & x_i^0\beta_1 & - & \dots & + & y_i & \cdot & y_i
\end{array}$$

Local reordering inside the addends lets us write the term the following way:

$$\begin{array}{ccccccccc}
 & x_i^0 x_i^0 & \beta_1\beta_1 & + & \dots & + & y_i x_i^0 & \beta_1(-1) \\
+ & x_i^0 x_i^1 & \beta_2\beta_1 & + & \dots & + & y_i x_i^1 & \beta_2(-1) \\
\vdots & & & & & & & \\
+ & x_i^0 x_i^{d-1} & \beta_d\beta_1 & + & \dots & + & y_i x_i^{d-1} & \beta_d(-1) \\
+ & x_i^0 y_i & (-1)\beta_1 & + & \dots & + & y_i y_i & (-1)(-1)
\end{array}$$

The left parts of the products (blue) correspond exactly to the elements of the matrix $C_i$ (the matrix $C$ from exactly one sample $i$), the right parts can be calculated by the matrix $D$.

$$D = \begin{bmatrix} \beta \\ -1 \end{bmatrix} \cdot \begin{bmatrix} \beta \\ -1 \end{bmatrix}^T = \begin{bmatrix} \beta_1\beta_1 & \dots & \beta_1(-1) \\ \beta_2\beta_1 & \dots & \beta_2(-1) \\ \vdots & \ddots & \vdots \\ \beta_d\beta_1 & \dots & \beta_d(-1) \\ (-1)\beta_1 & \dots & (-1)(-1) \end{bmatrix}$$

In conclusion, the RSS of a sample $i$, given the OLS model defined by $\beta$ in the matrix $D$, can be calculated by the grand sum of the Hadamard product $\odot$ (the element-wise product of two matrices of the same size) of $D$ and $C_i$.

$$RSS_i = \sum_{\ell,m=1}^{d+1} (D \odot C_i)_{\ell,m}$$

To calculate the overall RSS of a sample set with $n$ samples, we need to add the RSS of all individual samples together.

$$\begin{aligned}
RSS &= \sum_{i=1}^{n} RSS_i \\
&= \sum_{i=1}^{n} \sum_{\ell,m=1}^{d+1} (D \odot C_i)_{\ell m} \quad \bigg| \text{ factor in } \sum_{i=1}^{n} \\
&= \sum_{\ell,m=1}^{d+1} \left( D \odot \sum_{i=1}^{n} C_i \right)_{\ell m} \quad \bigg| C = \sum_{i=1}^{n} C_i \\
&= \sum_{\ell,m=1}^{d+1} (D \odot C)_{\ell m}
\end{aligned}$$

Given any sample set defined by $C$ and an OLS model defined by the vector $\beta$ in $D$, we can calculate the RSS of the samples in $C$ with the great sum of the matrix $D \odot C$. Dividing this value by the number of samples $n$ results in the MSE.

## B. Multidimensional Evaluation

This section evaluates the runtime and accuracy of our approach relative to the parameter $d$ (the number of input dimensions). The behavior in relation to parameter $d$ is *not* a fundamental novelty of our approach and therefore not the main contribution of our paper. All other mentioned approaches share the runtime complexity of $\mathcal{O}(d^2)$ with our approach.

At the same time, there are multiple key insights that are noteworthy to share. As representative real data for multidimensional datasets is hard to find and obtain, this analysis focuses on synthetic data. The general setup resembles the experiments done in Section 6.1 of the main paper.

### B.1. Discussion of the Sherman-Morrison Formula

For the evaluation with $d = 2$, all relevant matrices are at most symmetric matrices of dimension $3 \times 3$, the matrix $A$, which must be inverted—or decomposed and solved for a specific system of equations—is a symmetric matrix of dimension $2 \times 2$. The specific approach to do these operations has a negligible impact on the overall runtime of the algorithm. This drastically changes when scaling up $d$.

RUNTIME COMPLEXITY

To achieve a runtime complexity of $\mathcal{O}(d^2)$, the exact dynamic program (DP) and the last optimization step of our approach rely on precomputing the inverse of matrix $A$ and updating it when adding or subtracting one sample. The naïve way of inverting a square matrix would result in a runtime of $\mathcal{O}(d^3)$. While there are approaches to reduce this runtime complexity for large matrices, no known method reduces it to $\mathcal{O}(d^2)$. The update of an inverted matrix with one additional sample is also called a *rank-1 update* (R1U) and can be done more efficiently than a matrix inversion, using the Sherman-Morrison formula. However, this requires special attention in the detailed implementation.

Given the matrix $A$ with a precomputed matrix $A^{-1}$, and the vectors $u$ and $v$, the updated inverse value of $A + uv^T$ can be computed as follows[3]:

$$(A + uv^T)^{-1} = A^{-1} - \frac{A^{-1}uv^T A^{-1}}{1 + v^T A^{-1} u}$$

Directly using the above formula typically results in a trivial evaluation from left to right in the individual subterms. In case of the term $A^{-1}uv^T A^{-1}$ in the dividend, this is detrimental, since it results in a multiplication of two matrices of size $d \times d$, which cannot be computed in quadratic

---

[3]The vectors $u$ and $v$ can be different in the generic version of the formula, but in this scenario they are always the same vector, describing all independent values of a single sample. The matrix $A$ of an individual sample $i$ can be calculated by $A_i = uv^T = x_i x_i^T$.

time complexity. This can be solved using the following equation:

$$(A + uv^T)^{-1} = A^{-1} - \frac{(A^{-1}u) \cdot (v^T A^{-1})}{1 + v^T A^{-1} u}$$

Since matrix multiplication (including matrix-vector multiplication) is an associative operation, the above formula is mathematically identical to the first formula, but it consists of two matrix-vector multiplications and one final vector-vector multiplication. All these operations are trivially computable in quadratic time complexity.

Although the optimized rank-1 update is widely recognized, this particular implementation detail is easy to miss. For example, Acharya et al.'s DP algorithm overlooks this aspect—likely an unintentional oversight that could affect both accuracy and performance of the baseline algorithm, though its impact is minimal for $d = 2$ (the primary case evaluated in their work). Despite being theoretically quadratic in time complexity, their implementation does not achieve such efficiency.

ACCURACY

Using the Sherman-Morrison formula may raise concerns regarding numerical stability. Continuously updating $A^{-1}$ instead of recalculating it from all available data can lead to a gradual buildup of rounding errors over time. Although this issue appears less severe for *well-conditioned* matrices, even these cases have shown signs of reduced stability.

In our evaluation, we used 64-bit fixed-size floating-point numbers—commonly known as *doubles*—for the computations. This approach aligns with standard practices in similar calculations, though it can introduce the rounding errors mentioned earlier. Although a detailed analysis of the Sherman-Morrison formula's outcomes is beyond the scope of this paper, we provide several anecdotal insights based on our specific use case.

**Low Impact on Well-Conditioned Data:** In the final stage of our algorithm, we ensure that nearly all samples come from a single segment, yielding a regression function $\hat{f}$ that closely approximates the true underlying function—with only a few outliers misassigned during the initial segment placement. Under these conditions, we assume that our matrices are well-conditioned. Employing the Sherman-Morrison rank-1 update never lead to any noticeable drop in accuracy—even when dealing with very high dimensions $d$ or large numbers of samples $n$. This is clearly demonstrated by comparing the performance of our algorithm both with and without the rank-1 update.

**Well-Conditioned Data Reduces the Impact:** The DP often calculates the regression function across multiple segments (e.g., over the whole sample set). Using the optimized

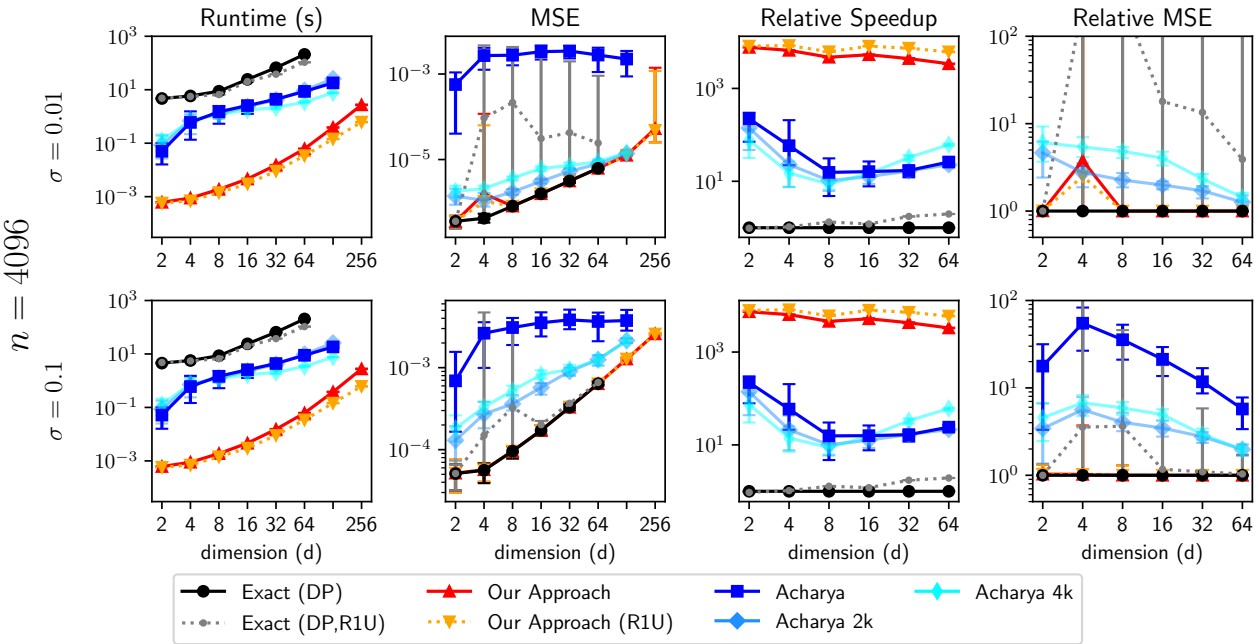

*Figure 6.* Varying number of arbitrary input dimensions with a constant number of samples. The dotted lines mark versions of the algorithms using the Sherman-Morrison formula for a quick rank-1 update (R1U). An outlier at $\sigma = 0.01, d = 4$ caused a high mean value, which is accentuated by the log-scale.

rank-1 update, resulted in a significantly higher error for DP but not for our algorithm. For $n = 8192$, the error was sometimes large enough to let the matrix seem like a singular matrix, which cannot be inverted at all. We assume this is due to the added samples significantly changing the current inverse. We observed no such issues with our algorithm— even at much larger values for $n$ and $d$—indicating minimal accuracy loss on well-conditioned matrices.

**Worse Accuracy with Trivial Implementation:** As mentioned above, Acharya et al.'s original implementation of the rank-1 update suffers more than necessary in terms of speed for large values of $d$. At the same time, this implementation also seems to yield much worse accuracy. Although these results are not shown here, they can be reproduced using the code provided in the supplemental material.

To ensure a fair comparison between the algorithms, we employ two modified variants of the dynamic program based on Acharya et al.'s implementation. The first variant, *DP*, avoids using the rank-1 update by solving all linear equations explicitly—this guarantees high accuracy at the expense of slower execution times. In contrast, the fast variant, *DP (R1U)*, incorporates fast rank-1 update described above to achieve quadratic time complexity while accepting a slight reduction in accuracy. We also present results for our algorithm with and without the rank-1 update, demonstrating that its performance remains robust under these different configurations.

## B.2. Fixed Number of Samples

Our experiment setup resembles the synthetic evaluation in the main paper. We measure both accuracy and runtime across 100 randomly generated piecewise functions. Accuracy is quantified using the mean squared error (MSE) between the estimated regression function $\hat{f}$ and the true underlying function $f$ at the positions of the input samples. The samples are uniformly distributed and the breakpoints (BP) are selected at random.

Unlike the original evaluation, we do not fit polynomials; instead, we use input data from multiple truly independent dimensions. For a fair comparison, it is still necessary that $n \gg d \cdot k$, because only one reasonable solution exists at $n = d \cdot k$. Since computational time is a limiting factor in our evaluation—especially when using DP—we reduced the number of breakpoints to $k = 4$, set $n$ to a fixed value, and varied parameter $d$.

Results for $n = 4096$ are presented in Figure 6. The figure demonstrates that, on occasion, our algorithm produces a suboptimal solution—an effect that is accentuated when the mean value is plotted on a log-log scale. In contrast, plotting the median aligns perfectly with the baseline curve for DP. Additionally, the rank-1 update significantly accelerates both DP and our algorithm for large values of d; however, unlike our approach, DP suffers from reduced accuracy when using the rank-1 update. The results also confirm that our algorithm consistently achieves higher accuracy than

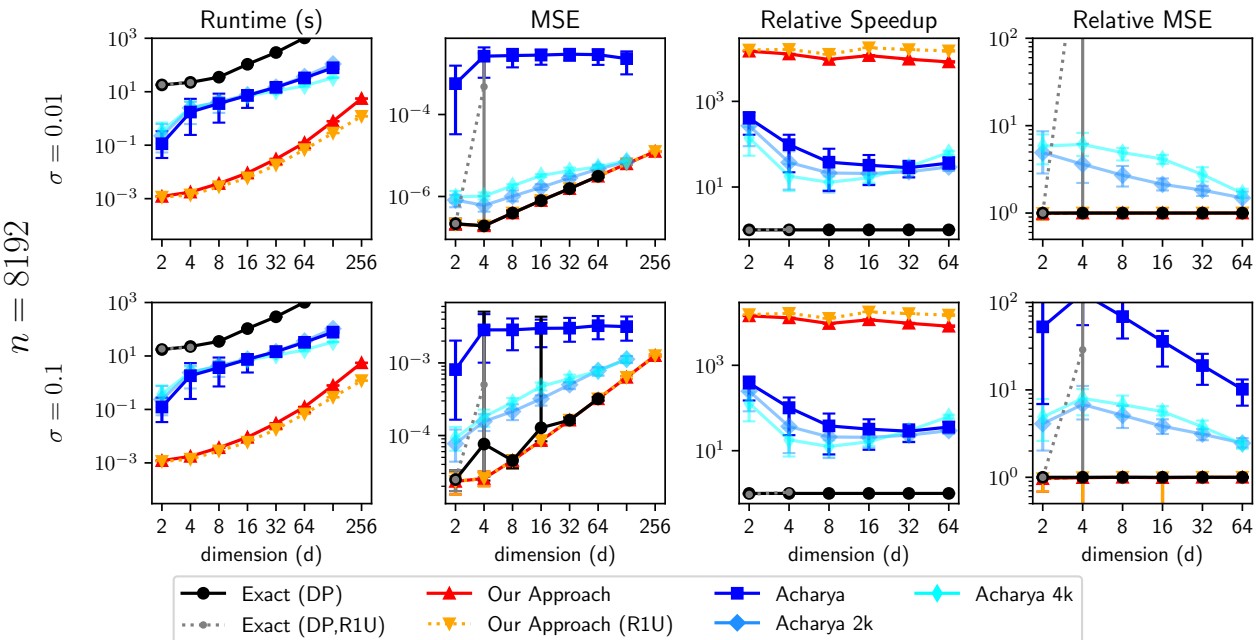

*Figure 7.* Similar experiment to the one shown in Figure 6, but using 8192 samples. The DP using the rank-1 update formula failed at $d = 2$ due to rounding errors.

the solution proposed by Acharya et al.—especially when exactly k segments are placed—and remains faster overall, although the gap in both performance metrics narrows as $d$ increases.

The primary reason behind Acharya et al.'s improved performance is the narrowing solution space as $d$ increases while $n$ remains fixed. A valid solution requires each segment to have at least $d$ samples. For example, with $d = 256$, $n = 4096$, and $k = 4$, there are only four times the minimum number of required samples. However, when considering Archarya et al.'s variant that places four times the segments (effectively resulting in $k = 16$), the method is forced into a scenario where only one viable solution exists—namely, an even distribution of breakpoints across the samples. This not only explains why Acharya et al. are closing the performance gap in terms of both speed and accuracy, but it also accounts for why at high $d$ values the $4k$ version outperforms the $2k$ version due to its more rapid collapse of the solution space.

To further validate our analysis, we conducted the same experiment using $n = 8192$. The results are shown in Figure 7. In this trial, the rank-1 update version of DP encountered early failure at $d = 4$ because the accumulated rounding errors resulted in a matrix that behaved like a singular one, i.e., a matrix that has a determinant of 0, which cannot be inverted. Although the naïve DP variant did not fail completely, it too struggled with accuracy issues at $d = 4$ and $d = 16$ with $\sigma = 0.1$. In contrast, our algorithm remained unaffected by these problems.

Observations indicate that the performance gap compared to Acharya et al. remains significantly larger and narrows later. This trend is particularly evident in the relative performance graphs. The impact of a collapsing solution space becomes especially prominent when comparing the runtimes of the 2k and 4k variants of Acharya et al. For $n = 4096$, the 4k variant starts outperforming the 2k variant just beyond $d = 8$; whereas for $n = 8192$, this advantage emerges between $d = 16$ and $d = 32$.

### B.3. Variable Number of Samples

Due to the algorithm's runtimes, increasing $n$ arbitrarily is not tractable. To keep the solution space size fixed, we adjust by scaling $n$ relative to $d$ instead of maintaining a constant number of input samples. Specifically, we set $n = 64 \cdot d$, which ensures that the solution space remains constant while keeping the DP runtime manageable. The results are shown in Figure 8.

The effects of employing a rank-1 update remain evident in the runtimes of both DP and our approach—with its impact limited solely to DP's accuracy. Moreover, our method again is affected by an isolated outlier in two distinct scenarios, much like when $n = 4096$. In all other cases, however, our accuracy aligns closely with that of the optimal baseline.

RUNTIME

For small values of $d$, the gap relative to Acharya et al.'s results is smaller; however, this may be due to measurement

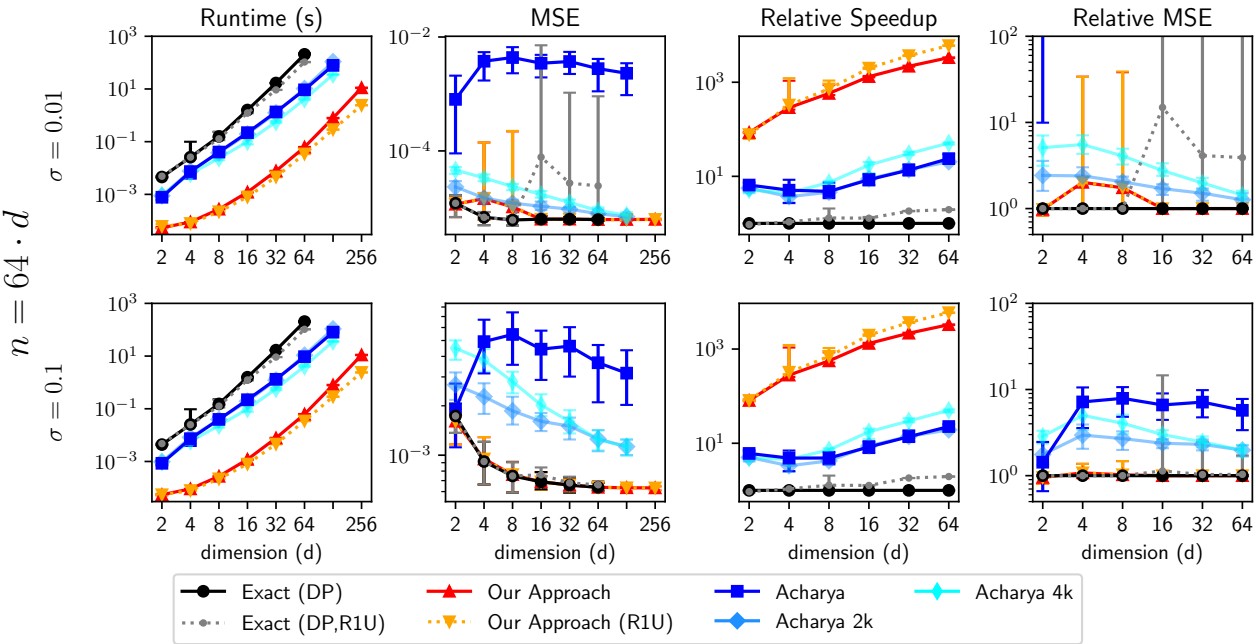

*Figure 8.* Similar experiment to the ones shown in Figure 6 and Figure 7, but this time the number of samples is not fixed. It is always 64 times the number of dimensions.

overhead since our timings were under one-tenth of a millisecond. For larger values, there are two key observations.

From $d = 8$ onward, the time for solving the linear equations begins to increasingly influence overall runtime. The version without the rank-1 update takes longer and approaches the runtime of Acharya et al., even though the runtime to break even is very large, probably somewhere between $10^4$ and $10^5$ seconds.

Incorporating the rank-1 update into our algorithm yields a constant speedup relative to Acharya et al.'s method, which is demonstrated by the parallel lines in the log-log runtime graphs. This is expected given that both approaches share identical asymptotic complexity with respect to parameters $d$ and $n$. In this specific configuration, we observe an approximate $200\times$ speedup, depending on the exact version.

ACCURACY

Measuring accuracy proves challenging and appears to be influenced by the noise variance present in the input data. The 2k and 4k variants of Acharya et al.'s method seem to converge toward the baseline as more samples and dimensions are added—this observation holds at least for $\sigma = 0.01$. This behavior may be linked to what is commonly known as the *curse of dimensionality*, where the space defined by increasing sample sizes expands much faster than linearly, resulting in a sparse representation of that space. Unlike experiments conducted with a fixed $d$, the baseline error does not decrease continuously with additional samples but

instead tends to stabilize at a near-constant value. Consequently, even a slight misalignment in the breakpoint has a diminishing impact as the number of samples used for calculating the MSE continues to grow.

The relative error plot does not clearly show that Acharya et al.'s version that places exactly $k$ segments begins to narrow the performance gap when both $d$ and $n$ increase. This uncertainty is especially pronounced for $\sigma = 0.01$.

### B.4. Conclusion

In evaluations using a fixed-size solution space, we consistently observe a constant speedup factor in runtime. As $d$ and $n$ increase, accuracy tends to level off—likely due to limitations in the experimental design or challenges with high-dimensional data, especially since improvements in the optimal baseline accuracy have plateaued.

Except for some rare outliers, our method is on par with the optimal solution and is significantly more accurate than Acharya et al.—especially compared to their version using exactly k segments. It is also faster than all other algorithms, though its speed advantage relative to Acharya et al. remains consistent.

Our main paper showed that using more samples increases the gap in speed and accuracy between our algorithm and others. We expect a similar trend if we increase the ratio of $n$ to $d$ beyond 64. A much lower value is not suitable in practice, since it approaches the minimum amount of needed samples and probably results in overfitting.

