# OpenReview forum: "Fast Min-$\epsilon$ Segmented Regression using Constant-Time Segment Merging"
_ICML.cc/2025/Conference — ICML 2025 poster_

### Official Review · Reviewer_Tjsb · 2025-03-02

**Overall Recommendation:** 3

**Summary:**

This paper provides a heuristic method to compute segmented regression. Instead of looking for the best segments directly, the algorithm finds as many segments as possible and then merges them until only k segments are left. The authors evaluate the algorithm on the synthetic datasets. The authors' method shows runtime or performance improvement compared to previous methods.

**Claims And Evidence:**

The authors' claims are proved by runtime analysis and experimental results.

**Essential References Not Discussed:**

I think it would be nice for the authors to discuss a bit about the relation between segmented regression and istronic regression in the literature review.

**Experimental Designs Or Analyses:**

N/A

**Methods And Evaluation Criteria:**

The MSE used in the evaluation is very standard.

**Other Comments Or Suggestions:**

N/A

**Other Strengths And Weaknesses:**

I am wondering if the authors can provide any theoretical guarantee for accuracy. I think this would definitely make the paper more interesting.

**Questions For Authors:**

I am wondering if the authors can mention a bit more about why segmented regression is an interesting problem, like what application it has.

**Relation To Broader Scientific Literature:**

I think this paper proposed an interesing heuristic method for segmented regressiion

**Theoretical Claims:**

The runtime analysis looks good to me.

---

> ### Author Rebuttal · Authors · 2025-04-01
>
> Thank you for reviewing our work, for the constructive improvement ideas and for pointing out the interesting direction of isotonic regression.
>
> **Relation to isotonic regression** (we assume that the review refers to isotonic regression): While isotonic regression is also based on an ordered sample set, it results in a predicted increasing constant per sample, using linear interpolation between the sample positions. The resulting regression is a (not necessarily strict) monotonically increasing piecewise linear function, where the number of pieces (i.e. segments) can be very high ($k \le n$). This problem setting is quite different to us, due to (a) the dynamic number of segments, (b) the enforced continuity, (c) the limitation to constant (or linear) models and (d) the restriction to monotonically increasing regressions. While the relation of isotonic regression to our approach is very interesting, we consider it to be a different type of problem.
>
> **Theoretical guarantees**: Indeed, based on the the way our algorithm works, we expect the accuracy guarantees of Acharya et al. (2016) to also hold true for our algorithm. At the same time, it is only possible to prove these statements for data with a very specific noise distribution. We instead focus on experiments with known distributions and real-world data with very unusual attributes to show the accuracy of our algorithm relative to the competing approaches.
>
> **Applications of segmented regression**: Despite the use cases listed in our introduction (ranging from ecology to econometrics to clinical guidelines), multiple approaches use these regressions for more efficient data structures [2,3]. A Nature Methods paper from this year [1] uses segmented regression as a step to model genetic data of tissue slices. In this example, it is not only necessary to have a small MSE, but to be able to get valuable breakpoint positions.
>
> We see regression in general as a fundamental building block in statistical analysis and machine learning. This can also be seen in the work of Diakonikolas et al. (2020), where segmented regression (Acharya et al. (2016)) was used to present a more efficient alternative to the CART-algorithm.
>
> [1] Chitra, Uthsav, et al. "Mapping the topography of spatial gene expression with interpretable deep learning." Nature Methods (2025): 1-12.
>
> [2] Galakatos, Alex, et al. "FITing-Tree: A data-aware index structure." Proceedings of the 2019 international conference on management of data. 2019.
>
> [3] Dai, Yifan, et al. "From WiscKey to bourbon: A learned index for Log-Structured merge trees." 14th USENIX Symposium on Operating Systems Design and Implementation (OSDI 20). 2020.

---

### Official Review · Reviewer_Qyh8 · 2025-03-10

**Overall Recommendation:** 3

**Summary:**

The paper addresses min-epsilon segmented regression, where the goal is to minimize the mean squared error (MSE) for a given number of segments. While the optimal solution has O(n^2)complexity (Bai & Perron, 1998), heuristics like Acharya et al. (2016) improve efficiency to O(n) but often introduce significant errors. The authors propose a method that merges segments using precomputed matrices, achieving 1,000 times lower MSE and 100 times faster runtime on large datasets. While promising, further clarification on theoretical guarantees and comparisons with recent heuristics would strengthen the evaluation.

**Claims And Evidence:**

The claims are supported by clear and convincing evidence.

**Essential References Not Discussed:**

N/A

**Experimental Designs Or Analyses:**

More comprehensive comparision results with the exisintg method shoud be provided and discussed

**Methods And Evaluation Criteria:**

The proposed method and evaluation criteria make sense for the problem considered in the paper

**Other Comments Or Suggestions:**

Please see Weaknesses.

**Other Strengths And Weaknesses:**

Strengths

- The paper appears to be novel.
- The paper is well-organized.


Weaknesses

- A more comprehensive comparison with existing methods should be provided and discussed, as the current results are relatively weak.

- The authors state that the state-of-the-art (SOTA) method is Acharya et al. (2016), which seems unusual given that it was proposed nine years ago. Why have no more recent approaches been considered? Are there other baselines that could be included?

**Questions For Authors:**

N/A

**Relation To Broader Scientific Literature:**

N/A

**Theoretical Claims:**

I reviewed them, but I didn't check the details.

---

> ### Author Rebuttal · Authors · 2025-04-01
>
> Thank you for your feedback.
>
> **Theoretical guarantees:** We would like to highlight that contrary to the reviewer's summary, the approach by Acharya et al. (2016), for a fixed value of $d$, improves the runtime of the approach to $\mathcal{O}(n\log{n})$, not $\mathcal{O}(n)$. This can be seen in Table 1 of our paper, and is stated in Section 3 of the paper by Acharya et al. (2016).
>
> **Related work and state of the art:** We discuss multiple different approaches in our related work.
> As discussed there, we believe that it would not be fair to compare against those, because they solve related, yet different problems.
> This is clarified in further detail in our answer to another review (please see our answer to Reviewer iMfW).
>
> While the approach of Acharya et al. (2016) is used and extended to solve further problems (e.g., segmentation with multidimensional breakpoints), it denotes the state of the art for min-$\epsilon$ segmented regression, together with the DP baseline. This is further underlined by it still being used in recent applications, for instance in a work [1] on gene expression analysis published in Nature Methods, which employs segmented regression to distinguish between different cell types, which we will mention and cite in the camera-ready paper.
>
> [1] Chitra, Uthsav, et al. "Mapping the topography of spatial gene expression with interpretable deep learning." Nature Methods (2025): 1-12.

---

### Official Review · Reviewer_9p1E · 2025-03-14

**Overall Recommendation:** 3

**Summary:**

This paper proposes a new heuristic method for the $\\min$-$\\epsilon$ segmented regression problem. Some prior works propose two types of algorithms for this problem. One line of work (Bai & Perron, 1998; Yamamoto & Perron, 2013) gives optimal solutions for this problem with computational complexity $\\mathcal{O}(n^2)$, where $n$ is the number of samples. Another work (Acharya et al., 2016) focuses on the case where $n$ is large and provides a heuristic algorithm with computational complexity $\\mathcal{O}(n \\log n)$, though it can result in large errors. This paper proposes a new heuristic method that achieves: (1) computational complexity $\mathcal{O}(n \\log n)$, and (2) (empirically) comparable to the optimal solutions. This paper provides computational complexity analysis for the proposed method and experiments showing the effectiveness of the proposed methods in terms of solutions quality and running time compared to prior works.

**Claims And Evidence:**

Yes.

**Essential References Not Discussed:**

N/A

**Experimental Designs Or Analyses:**

Yes. This paper contains two main experiments: (1) one with synthetic data, where they generate piecewise continuous functions with the number of pieces $k = 6$, and create data sets of $n$ points using the generated functions plus Gaussian noises. The results of this experiment are shown in Figure 3. (2) Another experiment with real data of 43 hours of CPU usage, measured every two seconds, resulted in 70,607 samples. The results of this experiment are shown in Table 2 and Figure 5.

**Methods And Evaluation Criteria:**

Yes.

**Other Comments Or Suggestions:**

I reiterate that I am not an expert in this field. Though I do not underestimate the contribution of this paper, I believe that the contribution in the ML aspect is not enough for an ICML publication. Since the main contribution lies in the algorithmic/computational aspect, I believe this paper should be submitted to other venues focusing on those aspects (e.g., STOC, SODA, or (somewhat) lesser conference like ITCS, etc.), or venues for data-mining (ICDM), or a more generic one like AAAI. **Though my overall recommendation is Weak Accept, my opinion is actually borderline, and I believe that the AC and other reviewers will have better judgement than me.**

### Other minor comments

1. At the beginning of page 2, please use itemize for the first paragraph (since the two contribution paragraphs also use itemize).

**Other Strengths And Weaknesses:**

Firstly, I am not an expert in this field so my opinion should be taken lightly.

## Strengths
The proposed method for $\\min-\\epsilon$ $k$-segment regression is computationally efficient (comparable to the prior heuristic approach) and achieves good performance (shown empirically, comparable to the optimal solution). I also appreciate that the paper presentation is straightforward to follow.

## Weaknesses

To me, the contribution of this paper is somewhat niche, and the contribution in the "machine learning" aspect does not meet the bar of a top ML conference. Here are the reasons.

### The problem considered is (somewhat) niche
   1. This paper considers the problem of $\\min$-\$\epsilon$ $k$ segmented regression. Though dealing with multi-dimensional data, the algorithm first has to choose a single axis/coefficient of the input and then perform segmenting along that particular axis. I know that this is not an issue with some specific types of data (e.g., time series), but it restricts the application of this method.

  2. Many other methods can deal with this general case (e.g., classification and regression tree (CART), multivariate adaptive regression splines (MARS), or regression tree, to name but a few), which is segmenting using multiple axes. I know that those methods had their own limitations, but the point I am making here is that the setting this paper considered is just a sub-case of a much broader problem.

  3. Besides, even if one restricts themselves to choosing only one axis to perform segmenting, choosing the axis itself is also a challenge.

### The main contribution lies in the computational aspect; contribution in the machine learning aspect is limited
   1. As far as I understand, the general idea of performing redundant segmentations along an axis and then merging was introduced by Arycha et al. (2016). I know it is controversial, but I think that the novelty of this paper lies in introducing a smarter way of merging segments that lead to improved solution quality (empirically).

   2. Moreover, though claiming that this method works well on a large dataset, it is only in terms of the number of samples $n$. As said, there is another critical factor in their computation cost, which is $\mathcal{O}(nd^2)$, where $d$ is the dimension of the input. This can scale up to $\\mathcal{O}(nd^3)$ if using standard implementation for calculating the inversed matrix, which is bad for high-dimensional data. Maybe that is why the experiments demonstrated in this paper were only conducted with data with small $d$.

  3. Most importantly, the machine learning contribution of this paper is limited. For example, I want to see how the predicted (segmented) functions perform on unseen data or extrapolate to data that lies outside the interval. I know that this is the limitation of the considered problem itself, not necessarily of the proposed method, and it is just my taste.

**Questions For Authors:**

See above.

**Relation To Broader Scientific Literature:**

Some prior works propose two types of algorithms for this problem. One line of work (Bai & Perron, 1998; Yamamoto & Perron, 2013) gives optimal solutions for this problem with computational complexity $\\mathcal{O}(n^2)$, where $n$ is the number of samples. Another work (Acharya et al., 2016) focuses on the case where $n$ is large and provides a heuristic algorithm with computational complexity $\\mathcal{O}(n \\log n)$, though it can result in large errors. This paper proposes a new heuristic method that achieves: (1) computational complexity $\mathcal{O}(n \\log n)$, and (2) (empirically) comparable to the optimal solutions. This paper provides computational complexity analysis for the proposed method and experiments showing the effectiveness of the proposed methods in terms of solutions quality and running time compared to prior works.

**Theoretical Claims:**

Yes. However, the theoretical side of this paper is minimal.

---

> ### Author Rebuttal · Authors · 2025-04-01
>
> Thank you for reviewing our work and for the suggestions for improvement.
>
> **Broader subject relevance:** We consider the topic of regression to be a fundamental building block for statistical analysis and machine learning.
> As mentioned in Section 7, Diakonikolas et al. (2020) have shown that an approach for multidimensional breakpoints based on the algorithm published by Acharya et al. at ICML 2016 outperforms CART, even when using constant segment models (for fairness against CARTs limitations).
> This suggests that fundamentally improving the basic segmented regression approach can lead to an improvement for those other problems.
>
> **Contribution:** From our perspective, our algorithm first and foremost drastically increases accuracy, rather than aiming to reduce computational complexity.
> If, for a given use case, sufficient resources are available, it is already possible to use the dynamic program.
> Our evaluation shows that this is not favorable for large datasets, as the number of samples would need to be reduced.
> In this case, the heuristics perform better.
> While our solution seems to be faster than Acharya's solution, the main contribution is its much higher accuracy and the focus on finding the correct breakpoints of the underlying data distribution.
> Therefore this work presents the first suitable algorithm if a time complexity of $O(n^2)$ is not acceptable and the exact positions of the breakpoints are important.
>
> It is also important to mention that our algorithm does not scale with $\mathcal{O}(nd^3 + n\log{n})$.
> This is true only for the specific implementation used in the evaluation, and it was mentioned only for transparency reasons.
> As stated in Section 5, our algorithm scales with $\mathcal{O}(nd^2+n\log{n})$.
> This is at least as good as the runtime complexity of Acharya's heuristic (cf. $\mathcal{O}(nd^2\log{n})$ in Section 3 of their paper).
> The limitation regarding the number of dimensions is based on the OLS and affects all competing algorithms in the same way.
>
> The reasons for the small value of $d$ in our evaluation are that (a) linear segmented regression is one of the most common use cases and best to illustrate, (b) $d$ and $k$ are typically constants that are chosen to model the data, which do not change, and (c) this allows us to perform a comparable evaluation to Acharya et al. (2016).
>
> **Relevance for ICML conference:** As stated above, we consider regression as a fundamental aspect of machine learning, that can be used on its own, but is also a building block to efficiently solve other problems in the domain of machine learning (as shown by Diakonikolas et al.). This is further underlined as the most relevant state of the art approach, Acharya et al. (2016), was also published at ICML.

---

### Official Review · Reviewer_iMfW · 2025-03-14

**Overall Recommendation:** 5

**Summary:**

The authors present a new method and algorithm for min-$\epsilon$ segmented
regression. The main contributions are primarily algorithmic but also related
to software engineering, as the authors implement highly efficient programming
techniques to enhance their implementation. The greedy algorithm they propose
merges neighboring segments in constant time. The paper includes one real-data
example and several synthetic benchmarks that demonstrate the efficiency of the
method.

## Update After Rebuttal

I stand by my original, high, score of the paper. I think it is valuable and interesting work
and that the paper is well-written and showcases strong results for the method. The
authors have also clarified some of my initial minor concerns.

**Claims And Evidence:**

The claims are generally convincing and the results are strong. The
problem is well-motivated and the method is clearly explained.

I would like to offer a couple of constructive suggestions:

- On line 431, first column, you mention that your algorithm performs better
  than certain alternatives from R and Python. However, I don't see evidence of
  direct comparisons to these alternatives in the paper. If such comparisons were
  conducted, including these experimental results would strengthen your claims.
  If not, it might be helpful to clarify this statement and perhaps explain why
  these comparisons weren't included. **Edit: I understand that these might not be
  warranted after all, so please disregard this comment.**
- It would be valuable to see an analysis of how well the theoretical
  complexity of your method is reflected in practical performance.

**Essential References Not Discussed:**

None that I could identify.

**Experimental Designs Or Analyses:**

The experiments are well-designed and the results are convincing.

**Methods And Evaluation Criteria:**

The benchmarks presented are appropriate for the problem at hand. Additional
benchmarks would further strengthen the paper, and perhaps the supplementary
material could be utilized to provide these.

**Other Comments Or Suggestions:**

- Figure 5 is large (in file size) and could benefit from being rasterized to
  improve loading times of the PDF.
- On line 307, second column, you suggest that results should not change by
  orders of magnitude due to the implementation language. While this is likely
  true when comparing Julia to C++, the statement may not hold for all language
  comparisons. A slight rephrasing might more accurately reflect this nuance.

**Other Strengths And Weaknesses:**

The paper is well-written and easy to follow, with a clear structure and
helpful illustrations that aid the reader in understanding the method. The
plots are well-designed and clearly convey the relevant information.

**Questions For Authors:**

No

**Relation To Broader Scientific Literature:**

The relevant preceding papers appear to be appropriately cited and discussed.
Table 1 effectively summarizes previous related contributions. The contribution
of the current paper is straightforward, aiming to improve complexity and
practical performance of this method.

**Theoretical Claims:**

I briefly checked the complexity analysis and could not find any issues.

---

> ### Author Rebuttal · Authors · 2025-04-01
>
> Thank you very much for your valuable and constructive feedback regarding our paper, including the code and experiment setting.
>
> **Evaluation design and constraints:** The alternatives in our 'related work' section solve a slightly different problem, e.g., by enforcing continuity of the resulting piecewise function.
> While we did not conduct a full evaluation, we first tried to analyze the shown real-world data using these algorithms, but the runtimes were intractably high for our use case (5-10 minutes for pwlf, compared to the sub-second runtimes of the presented heuristics, see Table 2).
> At the same time, the results were worse than the regression functions from the Acharya et al. heuristic, sometimes aborting because they did not converge to a solution at all.
> Directly comparing these runtimes or errors to our work in the evaluation seemed unfair, since the additional constraints change the problem.
> For the use case considered here, it does not make sense to choose these algorithms over any of the evaluated ones. We will rephrase the corresponding paragraph to express this setting more clearly.
>
> **Figure 5 and minor clarification:** We appreciate the additional feedback and the constructive suggestions to improve our work very much.
> Our goal was to plot the dataset in Figure 5 as accurately as possible. Still, we will change this by rasterizing the data or the whole image since the raw data is available digitally.
> Furthermore, we will definitely rephrase our statement regarding the programming language performance. It was meant with regards to the rest of the paragraph, i.e., in the context of our evaluation. We see that this sentence can be misleading if it is quoted separately.

---

> > ### Comment · Reviewer_iMfW · 2025-04-01
> >
> > Thanks for the reply! I will make some minor updates to the review regarding related work. I am somewhat disappointed, however, that you haven't indulged any of the reviewer's requests (mine included) for additional experiments. I think reviewer 9p1E raises some valid concerns as well regarding this point and the lack of investigation with respect to the dimension $d$. I have trouble seeing why *every* comparison would need to be comparable with respect to Arycha et al. (2016). Could you not just include an experiment to investigate the performance of your method by itself?

---

> > > ### Author Response · Authors · 2025-04-08
> > >
> > > Thank you for the elaboration. It was not our intention to ignore additional evaluations settings.
> > >
> > > Our main concern in the answer to reviewer 9p1E was to clarify the theoretical runtime complexity of our solution.
> > > Of course, equal (or even better) runtime complexity does not necessarily result in better practical compute performance for the chosen parameters.
> > > We did not consider $d$ as the most important metric in our evaluation strategy since it needs to be much smaller than $n$ anyway, and the main advantage over the competing accurate solution is the runtime relative to the amount of samples.
> > > Still, we do see that evaluations regarding parameter $d$ are also relevant, even if it is a parameter that is chosen when deciding on the modeling function.
> > >
> > > **Evaluation Update:** In the meantime, we made small changes to the already supplied source code and evaluated the runtime relative to parameter $d$ in the range $[2..256]$.
> > > The experiment aligns with Figure 3 of our paper; we generated 100 random curves for every setting with a specific number of dimensions $d$ and $k=4$.
> > > In terms of accuracy, our algorithm is still on par with DP and outperforms Acharya et al., despite not using more segments and not using knowledge about the noise distribution in the data.
> > >
> > > We believe that this further strengthens our claims on the performance and accuracy of our algorithm and we also hope that it increases the value proposition for the ICML community.
> > > We will gladly share this analysis with a more detailed explanation supplementary to our paper (together with the updated code and data).
> > > A preliminary version of the evaluation figure is available at:
> > >
> > > https://icml-segreg-fig-eval-dim.tiiny.site/
> > >
> > > **Notes on $n$ when Evaluating the Parameter $d$:**
> > > It is important to note that using a fixed $n$ is somewhat unrepresentative at high values of $d$.
> > > Given that Acharya 4k is placing up to 16 segments in this case, at $d=256$ and $n=4096$, there would only be one plausible solution left (16 segments just barely fit, as no segment model should be underdefined, so the result is a perfect (over-)fit).
> > > That is, the solution space - the size of the set of sane breakpoint positions - is drastically reduced for high values of $d$, and there is nothing really left to decide or optimize, resulting in increasingly similar runtimes.
> > > This can be seen in the figure (see above): The relative speedup of Acharya starts to increase again at higher values for $d$ for constant $n$, and the increase in speedup is much weaker for $n=8192$ than for $n=4096$.
> > >
> > > In another experiment, we set $n = 64 \cdot d$.
> > > This prevents the solution space from collapsing as we scale up $d$.
> > > Since Acharya's approach scales similar to our solution in terms of $n$, we can thereby analyze the scaling in terms of $d$.
> > > The analysis shows that we are always faster in the evaluated range, but start to scale worse for $d \ge 32$ without the Sherman–Morrison rank-1 update (R1U) when the cost of $d$ starts to dominate the runtime.
> > > This corresponds to our theoretical analysis, which states the $\mathcal{O}(d^3)$ runtime for our old implementation.
> > > With R1U as described in the paper results in a parallel line to Acharya et al., which indicates that both algorithms scale identically (with $\mathcal{O}(d^2)$ according to our complexity analysis as well as Acharya et al.'s) when $d$ starts to dominate the runtime cost (compared to the number of samples $\mathcal{O}(n\log{n})$).
> > > This corroborates our theoretical analysis of the computation time relative to Acharya et al.'s.
> > >
> > > **Notes on the Implementation of DP:**
> > > The DP relies heavily on precomputing an inverse matrix and then using R1U to achieve a time complexity of $\mathcal{O}(d^2)$.
> > > Acharya et al.'s implementation of the DP sometimes struggles with accuracy for higher $d$, even failing early on in case of $n=8192$.
> > > We made minimal modifications to directly solve the linear equations instead of using the inverse matrix to have a reasonable baseline in these settings.
> > > While this results in a theoretical time complexity of $\mathcal{O}(d^3)$, this did not change the practical runtime in a substantial way for DP.
> > > Using R1U on our algorithm does not impact accuracy, but it does reduce the runtime, especially for $d \ge 32$.

---

### Decision · Program_Chairs · 2025-05-01

**Decision:**

Accept (poster)

**Comment:**

Overall, the reviewers found that the paper is well-written, and a valuable and interesting work. However, they also raised some concerns, including about limited theoretical novelty and insufficient comparison with existing methods, which should be carefully addressed.